# Genome-Wide Identification of Wheat KNOX Gene Family and Functional Characterization of *TaKNOX14-D* in Plants

**DOI:** 10.3390/ijms232415918

**Published:** 2022-12-14

**Authors:** Song Li, Yaxin Yao, Wenjie Ye, Shaoyu Wang, Chao Zhang, Shudong Liu, Fengli Sun, Yajun Xi

**Affiliations:** College of Agronomy, Northwest A&F University, Xianyang 712100, China

**Keywords:** wheat, KNOX gene family, mechanical damage, cold stress, leaf development, *KNOX* expression pattern

## Abstract

The KNOX genes play important roles in maintaining SAM and regulating the development of plant leaves. However, the *TaKNOX* genes in wheat are still not well understood, especially their role in abiotic stress. In this study, a total of 36 KNOX genes were identified, and we demonstrated the function of the *TaKNOX14-D* gene under mechanical injury and cold stress. Thirty-six *TaKNOX* genes were divided into two groups, and thirty-four *TaKNOX* genes were predicted to be located in the nucleus by Cell-PLoc. These genes contained five tandem duplications. Fifteen collinear gene pairs were exhibited in wheat and rice, one collinear gene pair was exhibited in wheat and Arabidopsis. The phylogenetic tree and motif analysis suggested that the *TaKNOX* gene appeared before C3 and C4 diverged. Gene structure showed that the numbers of exons and introns in *TaKNOX* gene are different. Wheat *TaKNOX* genes showed different expression patterns during the wheat growth phase, with seven *TaKNOX* genes being highly expressed in the whole growth period. These seven genes were also highly expressed in most tissues, and also responded to most abiotic stress. Eleven TaKNOX genes were up-regulated in the tillering node during the leaf regeneration period after mechanical damage. When treating the wheat with different hormones, the expression patterns of TaKNOX were changed, and results showed that ABA promoted *TaKNOX* expression and seven *TaKNOX* genes were up-regulated under cytokinin and auxin treatment. Overexpression of the *TaKNOX14-D* gene in *Arabidopsis* could increase the leaf size, plant height and seed size. This gene overexpression in *Arabidopsis* also increased the compensatory growth capacity after mechanical damage. Overexpression lines also showed high resistance to cold stress. This study provides a better understanding of the TaKNOX genes.

## 1. Introduction

The leaf base, leaf edge, leaf tip and leaf vein form the complete leaves [1]. The early stage of a leaf is the leaf primordia, which forms in the apical meristem. The procedure of leaf primordia developing into leaves is regulated by various genes, and these genes are co-expressed [1,2]. Among these genes, KNOXI is one important gene. This gene’s high expression (Knotted-like homeobox 1) is the basic feature of the shoot apical meristem (SAM) in maintaining the stem cell state [3]. Leaf initiation requires the collaboration of auxin, cytokinin and gibberellin. At the P0 site (the site of leaf primordia formation), when the concentration of auxin reaches the maximum value and gradually connects to the existing vasculature, this process with the cytokinin and gibberellin responds, and the leaf primordia develops into leaf [4,5,6]. However, in this process, KNOXI gene expression decreased [7]. Low expression of the KNOXI gene is a signal for leaf initiation in the SAM [7]. Previous studies have also shown that the KNOXI protein plays a crucial role in maintaining a transient morphogenetic window during early leaf development in some species [8,9]. In simple leaf species, the KNOXI gene is a key factor in regulating leaf shape, including junction, coiled or shallow cracked leaves, and the ectopic meristem [10].

In maize, the KNOX1 protein knotted-1 (*KN1*) expresses in the proximal region of the primordia firstly, then the expression of *KN1* transfers to the distal locations; this result suggests that *KN1* is involved in the establishment of proximal/distal polarity [11,12]. In Arabidopsis, the KNOXI gene includes *STM*, *KNAT1*, *KNAT2* and *KNAT6*. When *KNAT1* and *KNAT2* expression were inhibited by the genes of *AS1* (ASYMMETRIC LEAVES1) and *AS2* during leaf development, the leaf shape was normal, while the mutant of *as1* and *as2* showed a cleft leaf [9,13]. Some research showed that the expression level of *KNAT1* in leaf margin mustard is higher than in serrated leaf margin mustard, suggesting that *KNAT1* may be involved in leaf margin development. Likewise, inhibited KNOXI gene expression would accelerate leaf maturation and reduce leaf complexity [14,15]. Overall, the Arabidopsis leaves might evolve from a more complex leaf ancestor through the inhibition of KNOXI expression [16]. The KNOX gene family is a conserved gene family in the plant. Most KNOX proteins have four characteristic conserved domains: KNOX1, KNOX2, ELK and HOX [17,18]. The first KNOX gene was named *Knotted-1* (*KN1*) in maize. Subsequently, KNOX genes were identified in many plants and the function of some KNOX genes have been proven [19,20,21]. According to the expression patterns of the KNOX genes, the KNOX genes were divided into KNOXI, KNOXII and KNATM, where KNATM is unique in dicotyledons [17,18]. The KNOX genes have different expressions and biological functions. KNOXI are mainly expressed in the SAM, and they play an influential role in the differentiation and maintenance of the meristem [18,22]. In Arabidopsis, the KNOXI gene *STM* plays an important role in the establishment of the SAM in Arabidopsis embryos. Further research proved the Arabidopsis *STM* mutant seeds only produced cotyledons, and could not produce new leaves [10]. The KNOXI genes are also closely associated with callus differentiation in the transgenic process. In rice with the *osh1*/*osh15* double-deletion mutants, the callus only formed a leaf-like structure, without bud formation [21]. When transferring maize KNOX gene *KN1* into tobacco, the rate of genetic conversion into seedling increased compared to the control [23]. Likewise, the KNOXI genes also play a crucial role in leaf shape formation, internode elongation, hormone balance and inflorescence structure formation [10,24]. The KNOXII are also studied in many species and aspects, such as leaf development and seed physical dormancy [25,26]. The KNOXII genes are expressed in all plant tissues. Recently, a study found that KNOXII genes are also related to the formation of the secondary cell wall in Arabidopsis [27]. *KNAT7* and *KNAT3* are two KNOXII genes of Arabidopsis, which synergistically affected the deposition of secondary cell walls to improve the mechanical support of the Arabidopsis stem [27]. KNATM is a specific subfamily in dicotyledonous plants, which encodes a MEINOX domain but not a homeodomain [17]. The KNATM gene interacts with other members to modulate their activities [28]. Moreover, *KNATM* is expressed in proximal-lateral domains of organ primordia and at the boundary of mature organs; in accordance, genetic analyses identify a function for *KNATM* in leaf proximal-distal patterning [17].

Wheat is one of the most important crops in the world, and about 35~40% of the global population rely on this as their main diet [29]. Although Han has identified this gene family in wheat and analysed the function of *TaKNOX11-A* under salt stress and PEG6000, the function of KNOXII genes under mechanical damage and cold stress are still not well understood [30]. In this study, the KNOX genes’ chromosome distribution, gene replication, evolutionary relationship, gene structure, *cis*-acting elements and expression patterns of tissues were analyzed. Likewise, the KNOX gene function was verified by transferring *TaKNOX14-D* into Arabidopsis. These results set the stage for further studies on the function and regulatory mechanisms of the KNOX genes in wheat. It also provided a theoretical basis for cultivating double-purpose wheat.

## 2. Results

### 2.1. Identification of KNOX Genes in Wheat and Prediction of Encoded Proteins

Based on the HMM files of KNOX1/KNOX2, and using HMM search, a total of 36 KNOX genes were identified (34 KNOX genes were identified by Han, and 2 new genes were identified in this study). We named these genes according to Han, and two new KNOX genes were named *TaKNOX15* and *TaKNOX16* [30] (Appendix A). The KNOX genes’ distribution on the wheat chromosomes are not even, with sub-genome A having the most KNOX genes, up to 13, sub-genome B having 12 KNOX genes and sub-genome D having 11 genes. The sequences length of TaKNOX proteins ranged from 153 aa to 389 aa. The molecular weight values and isoelectric point values (pI) ranged from 5.15 to 9.11. Most proteins’ pI was less than 7, indicating that the TaKNOX protein was acidic. Subcellular localization predictions indicated that the majority of KNOX genes localized at the nucleus, and few genes localized at the chloroplasts and mitochondria.

### 2.2. Gene Duplication and Homology Analysis of the KNOX Gene Family

The KNOX genes distributed in chromosomes unevenly. Chromosome 4A had the most KNOX genes, containing 6 KNOX genes. Chromosomes 2, 6 and 7 had three genes, which distributed on A, B and D chromosomes, respectively. Most KNOX genes were located at the ends of chromosomes (Figure 1A). Gene duplication is a normal event in plant genomes [31]. In this study, 5 tandem duplications were found in 36 *TaKNOX* genes (Figure 1A, Appendix A). An additional 24 segment duplicates were found to be within the fragmentary collinearity block, including the majority of *TaKNOX* genes with partial homologous relationships. These results indicated the presence of tandem and fragment repeats in the *TaKNOX* gene expansion process in wheat.

The KNOX genes’ homology between wheat, rice, *Arabidopsis thaliana,* barley and maize were also analyzed, and the results showed that only 1 collinear gene pair existed between wheat and *Arabidopsis*, 13 collinear gene pairs existed between rice and wheat, 12 collinear gene pairs existed between barley and wheat and 18 collinear gene pairs existed between maize and wheat (Figure 1B, Appendix A). The results suggested that wheat has more collinear pairs with monocotyledon plants. 

### 2.3. Phylogenetic Tree and Motif Analysis of KNOX Genes in Multiple Species

To analyze the evolutionary relationship between wheat KNOX genes and other species, the KNOX genes in two C4 species (maize and sorghum), three dicotyledon species (*Arabidopsis*, tomato and soybean) and rice were identified. Meanwhile, 10 motifs of proteins were analyzed via the MEME (https://meme-suite.org/meme/) (Appendix A). The results showed that KNOX genes were identified in all C3 (wheat, rice, *Arabidopsis*, tomato and soybean) and C4 plants, while the number of KNOX genes varied greatly (Figure 2, Appendix A). Based on the sequence similarity of these KNOX genes, we divided the KNOX genes into three groups: class I, class II and class M. Class I contained the largest number of genes, up to 78 genes, and class M only contained 1 *Arabidopsis* gene (Figure 2). All gene proteins contained motif3. motif1/motif2 and motif4 were also found in most genes (Appendix A). In class I, all species of KNOX genes were contained. motif8 is a unique motif, which only existed in class I, and about 66.7% of KNOX genes were contained in this motif. In class M, the only Arabidopsis gene, KNATM, resided. KNATM was the shortest KNOX gene of all KNOX genes, and only one motif existed (Figure 2). There were no KNOX genes in soybean, and tomato was divided into this group, which suggested that KNATM was the unique gene in Arabidopsis. All species also contained class II KNOX genes, and motif10 is the unique motif for this group. In this group, 30 KNOX genes had this motif, while in 30 genes, only 1 Arabidopsis (*KNAT7*) contained motif10 (Figure 2). motif5 was also specific to class II, which was not identified in other groups (Figure 2) 

### 2.4. Gene Structure and Cis-Acting Elements of TaKNOX Genes

Gene structures were analyzed using Tbtools. According to the homology relationship of wheat KNOX genes, the KNOX gene family was divided into two groups: class I and class II (Figure 3A, Appendix A). The gene structures of the KNOX genes had significant differences between these two groups. In class I, most exons were assembled at the N-terminal of the gene and fewer at the C-terminal. The exon numbers of genes in class I were concordant. In contrast to class I, the exons of genes in class II showed opposite distributions. In class II, there were more exons assembled at the C-terminal and the exon numbers were inconsistent (Figure 3A). In class I, *TaKNOX7-A* was the longest KNOX gene, and this group also had the shortest gene, *TaKNOX5-A*. Gene structure showed genes with close phylogenetic relationships had similar gene structures (Figure 3A). 

The 2000 bp gene sequences before the CDS were downloaded as the promoter for analyzing the *cis*-acting elements [32]. In this study, ten *cis*-acting elements were identified with PlantCare, including six hormone response elements (MeJA, ABA, GA, ETH, zein and IAA). The results showed that almost all *TaKNOX* genes had MeJA-responsiveness, except *TaKNOX5-D* and *TaKNOX15* in class I (Appendix A). *TaKNOX2-A* had the maximum MeJA-responsiveness elements, up to 26 (Figure 3B,C). Abscisic acid responsiveness was also found in most KNOX genes. *TaKNOX8-D* had the largest abscisic acid responsiveness. Auxin-responsiveness elements were also found in most KNOX genes, especially in class II, and *TaKNOX13-B* had the most auxin-responsiveness (Figure 3B,C). Another three hormone responsiveness elements were also found in the KNOX genes. Compared to zein metabolism and ethylene-responsiveness elements, the gibberellin-responsiveness elements found in KNOX promoter were minor (Figure 3B,C). Only 11 KNOX promoter contained this *cis*-acting element, and it was mainly distributed in class I. The *cis*-acting element of CAT-box is related to meristem expression, and this element was found in both groups (Figure 3B,C). 

### 2.5. Network Construction of TaKNOX Cascade Genes

Based on miRNA and TaKNOX target relationships, a miRNA and *TaKNOX* gene co-expression regulatory network was conducted. In this experiment, approximately half of KNOX genes were targeted by 10 miRNAs (Figure 4, Appendix A). Based on the target relationship, 24 miRNA–TaKNOX interactions were constructed (Figure 4). tae-miR408 was targeted by the most genes, up to ten *TaKNOX* genes, followed by tae-miR160, which was targeted by three *TaKNOX* genes. The rest of the miRNA was only targeted by one TaKNOX gene. Cleavage was the main way that miRNA inhibited KNOX gene expression, which accounted for 77.8% (Appendix A). In addition, miRNAs were primarily targeted in the CDS region, but at different locations; for example, tae-miR160, which targeted the *TaKNOX* gene, was behind the KNOX domain, while tae-miR408, which targeted the *TaKNOX* gene, was before the KNOX domain, but they both silenced gene expression.

### 2.6. Expression Patterns of KNOX Genes in Different Growth Stages and Tissues and under Different Abiotic Stresses

Gene expression is spatiotemporal, and this gene family has been reportedly related to plant leaf development, thus, we analyzed the KNOX gene expression patterns in different fertility periods and different tissues. Genes under different abiotic stresses were also analyzed (Figure 5). The *TaKNOX* gene expression patterns of the wheat growth stage were downloaded from the Expvip (http://www.wheat-expression.com/). The results showed that the expression patterns were divided into three groups (Figure 5A). In class II, seven *TaKNOX* genes were expressed in whole growth periods. In class I-A, there were fourteen *TaKNOX* genes not expressed or expressed at low levels during the wheat growth period (Figure 5A). In class I-B, most genes were expressed at seeding, tillering, flag leaf, heading, anthesis and milk grain stages. Compared to other *TaKNOX* genes, *TaKNOX14-D* had the highest expression level, especially at the tillering stage. 

Tissue expression patterns were also analyzed in this study. As for the *TaKNOX* expression patterns at different growth periods, tissue expression patterns were also divided into three groups. In class I-A, eight genes were not expressed in all tissues. Six genes had different tissue expression profiles, four genes were expressed at low levels in the stem and three *TaKNOX* genes were expressed at low levels in the SAM and stamen. *TaKNOX2-B* had higher expression levels in the stamen and pistil compared to other genes in class I-A (Figure 5B). In class I-B, *TaKNOX* genes were mainly expressed in the SAM and stem. (Figure 5B). Three *TaKNOX* genes were highly expressed in the stamen compared to other genes in class I. In class II, seven *TaKNOX* genes were expressed in all tissues and highly expressed in leaves, awns and glumes (Figure 5B). *TaKNOX14-D* had the highest expression in leaves, spikes and awns (Figure 5B). 

In order to analyze the expression of *TaKNOX* genes under different abiotic stresses, data on expression levels under abiotic stress were also downloaded from the wheat expression web. The results showed that 36 *TaKNOX* genes had 4 expression patterns and were divided into 3 groups (Figure 5C). In class I-A, two expression patterns were included. The first expression pattern consists of three genes that were up-regulated under phosphorus starvation, while down-regulated at PEG6000 stress. These three genes were not expressed under other abiotic stresses (Figure 5C). The remaining 14 genes in class I-A were not expressed in all abiotic stress. In class I-B, all genes were only expressed in normal conditions (Figure 5C). Seven genes in class II had different expression patterns. *TaKNOX31* and *TaKNOX14-D* were highly expressed under phosphorus starvation, and *TaKNOX14-D* was also highly expressed under PEG6000 and cold stress. *TaKNOX35* and *TaKNOX36* were extremely expressed under cold stress (Figure 5C). These results suggested that the expression of the *TaKNOX* gene was correlated with the stage of wheat growth.

### 2.7. Expression Patterns of KNOX Genes under Different Hormone Treatments

For the promoter of *TaKNOX* genes containing different hormone response elements, the expression patterns of *TaKNOX* genes under different hormone treatments were analyzed (Appendix A). Results showed that *TaKNOX* genes had different expression patterns under hormone treatment, and about half of *TaKNOX* gene expression was inhibited (Figure 6, Appendix A). Based on the expression patterns of the *TaKNOX* genes, we divided these expression patterns into two groups: class I and class II. In class I, the *TaKNOX* genes were highly expressed under ABA treatment, especially the genes *TaKNOX5-B* and *TaKNOX10-B*. These two genes expression levels were more than 50 times higher than CK. Three genes (*TaKNOX1-B*, *TaKNOX2-A* and *TaKNOX3-B*) were also up-regulated under CTK, GA and IAA treatment, whereas other genes in class I were down-regulated under hormone treatment, except ABA (Figure 6). In class II, *TaKNOX* genes were also up-regulated under ABA treatment. Interestingly, the expression patterns in class II showed much difference. The expression of *TaKNOX12-B* was inhibited under five hormone treatments, while there were two *TaKNOX* genes (*TaKNOX14-D*, and *TaKNOX12-D*) up-regulated under the other five hormone treatments in class II. *TaKNOX11-A* and *TaKNOX9-D* were also up-regulated under most hormone treatments, except MeJA (Figure 6). 

### 2.8. Expression Pattern of KNOX Genes after Mowing

The KNOX genes are involved in leaf development, so we wondered if this gene family was involved in the regeneration of wheat leaves after mowing. As shown in Figure 7, these expression patterns were divided into two groups. In class I, *TaKNOX8-D* had the highest expression level after mowing in the tillering node (TN) after 24 h, and the expression pattern in the root tip (RT) was similar to the TN (Figure 7, Appendix A). The expression patterns of *TaKNOX4-B*, *TaKNOX6-D1* and *TaKNOX8*- were similar and were slightly up-regulated after mowing in TN, but in RT the expression patterns were different. *TaKNOX4-B* was down-regulated after mowing. *TaKNOX18* was down-regulated and then up-regulated, and the highest expression level in the RT was at 144 h after mowing. Meanwhile, *TaKNOX8-A* was slightly up-regulated at 0.5 h and 2 h and then down-regulated. (Figure 7). Additional *TaKNOX* genes, such as *TaKNOX1-B*, *TaKNOX2-A*, *TaKNOX3-B* and *TaKNOX4-B,* were highly expressed at 2 h and 8 h. *TaKNOX4-D* was down-regulated at 24 h, while up-regulated at other times in the TN. This gene had the highest expression level at 8 h in the RT. The remaining *TaKNOX* genes in class I were slightly up- or down-regulated in the TN. These genes were also slightly down-regulated in the RT. *TaKNOX4-A* expression in the RT was up-regulated after mowing, especially at 72 h (Figure 7). 

The expression patterns of *TaKNOX* genes in class II differed from class I. Genes in class II also showed different expression patterns. *TaKNOX11-A* and *TaKNOX12-B* were down-regulated firstly, and then up-regulated in the TN. The expression level of *TaKNOX12-D* decreased at 0.5 h, 24 h and 72 h, while it increased at 2 h and 144 h (Figure 7). The expression pattern of *TaKNOX* genes in the RT showed similarly. It was up-regulated firstly, and then down-regulated (Figure 7). *TaKNOX14-D* had the highest expression level at 24 h after mowing compared to other genes in class II. This gene was also highly expressed at 8 h (Figure 7). Results showed that mowing could change *TaKNOX* gene expression. 

### 2.9. Gene Function Verification in Arabidopsis thaliana 

The gene *TaKNOX14-D* was cloned and transferred into *Arabidopsis* for verification of the gene function. After the transgenic *Arabidopsis* was homozygous, three transgenic lines were randomly selected, named OE-1, OE-2 and OE-3 and were used for gene function analysis.

Three transgenic lines and WT were planted in MS media. After four *Arabidopsis* lines reached the 4-leaf stage, the root length and leaf width were counted. The results showed that the seedling root’s length of the transgenic lines was significantly longer than the WT, and the OE-3 had the longest roots (Figure 8A and Appendix A). The leaf width and length of three transgenic lines was also significantly larger than WT. OE-3 had the longest leaf width and length (Figure 8B,C). The results suggested that *TaKNOX14-D* overexpression in *Arabidopsis* could promote plant leaf and root growth.

Three transgenic lines’ growth rate was faster than WT in this experiment. These three lines reached the bolting period, flowering period and pod period earlier than WT (Figure 8D,G,J). After the transgenic lines reached the bolting-flowering period, the leaf width and length of *Arabidopsis* was counted again, and the results showed that the leaf width and length were still significantly larger than WT, especially OE-3, which was nearly twice more than that of WT (Figure 8E,F). Moreover, when the four lines of *Arabidopsis* reached the bolting stage, the number of rosette leaves in the transgenic lines were significantly more than WT, by about four leaves (Figure 8E,F). OE-3 had the most rosette leaves, and OE-2 had the least. Subsequently, the chlorophyll content of rosette leaves of *Arabidopsis* was detected at different growth periods. The results showed that the chlorophyll content of four *Arabidopsis* lines increased gradually with the growth period extension (Figure 8H). The chlorophyll content of three transgenic lines were significantly higher than WT at three growth periods (Figure 8H). Interestingly, OE-3 had the longest leaf width and length, but the chlorophyll content was the lowest among three transgenic lines. 

The leaf width and length of transgenic lines was longer than WT. A GUS staining assay was used to prove this phenotype. The roots, stems, leaves, flowers and pods of four lines were collected and photographed after GUS staining. As shown in Figure 8I, the leaves and flowers of transgenic lines were dyed, while WT was not. Moreover, the young leaves were dyed deeper than old leaves, which indicated *TaKNOX14-D* had higher expression levels in young leaves than in old leaves (Figure 8I). Other tissues were not dyed, which was the same as WT (Figure 8I and Appendix A). We also detected the expression levels of the genes associated with leaf growth, including the *AUX* and *IPT* genes. The results showed that the *IPT* gene expression level was increased in transgenic lines, while the *AUX* gene was slightly down-regulated (Appendix A). These results suggested that cytokinin content might increase in the transgenic lines compared to WT, while auxin content might not change. Overexpression of the *TaKNOX14-D* gene also had an effect on *Arabidopsis* height (Figure 8J,K). The transgenic line’s height was nearly 2-fold higher than WT under normal growth. The gene *GA20ox* was higher expressed in transgenic lines than in WT, this might account for the height of transgenic lines being higher than WT (Appendix A). The pod lengths of the four lines were also compared after *Arabidopsis* matured. The result showed that the pod length of transgenic lines was significantly longer than the wild-type lines, and the OE-3 line had the longest pods (Figure 8L,M). The seed volumes of three transgenic lines were also bigger than WT. OE-1 seed’s volume were the biggest, which was reflected in the width and length. OE-2 and OE-3 *Arabidopsis* seeds were slightly smaller than OE-1, but were still significantly bigger than wild-type (Figure 8N,O).

### 2.10. Gene Function Analysis by Arabidopsis thaliana under Mechanical Damage and Cold Stress

When *Arabidopsis thaliana* was at the bolting-flowering stage, the stem was clipped for detection of the compensatory growth ability of four lines. In this study, the time that these four lines reached the bolting-flowering stage after clip treatment, Fv/Fm, chlorophyll contents and pod length were counted. Transgenic lines had better compensatory growth capacity after mechanical damage, for they used shorter times to reach the bolting-flowering period again (Figure 9A,B). Four days were used by the three transgenic lines to reach the shoot-flowering stage, while seven days were used by the WT. The chlorophyll content of rosette leaves was detected in four lines before mechanical damage, and the content was detected again after the four lines reached bolting-flowering stage. The result showed that mechanical damage caused the reduction of chlorophyll content in the rosette in four lines, but the reduction level of WT was the maximum (Figure 9C). This result also showed that the chlorophyll content of three transgenic lines was significantly higher than WT after mechanical damage (Figure 9C). The chlorophyll fluorescence showed that the PSII maximum photochemical efficiency (Fv/Fm) of the transgenic lines was significantly higher than that of WT (Figure 9D).

Interestingly, the plant height of transgenic lines was similar to WT after mechanical damage, which differed from the plant height under normal growth conditions (Figure 9E). However, the aboveground biomass of transgenic lines was significantly more than WT. As showed in Figure 9E (red box), the stem number of three transgenic lines were more than 6 stems greater than that of WT (Figure 9F). Mechanical damage could promote pod growth, for the length of pods was increased after clipping compared to the length of pods under normal conditions (Figure 9G,H). As we expected, the pod length of three transgenic lines was still longer than WT (Figure 9G,H). The seed volumes of three transgenic lines had no significant difference from the WT, though the OE-1 seed widths were a bit longer than WT (Appendix A). The seed widths of OE-3 were also longer than WT, but the seed lengths were shorter than WT (Appendix A).

Because the wheat *TaKNOX14-D* gene was highly expressed under cold stress, the *Arabidopsis* resistance to cold was also identified. The transgenic lines and WT were planted in MS medium and treated with cold for seven days. The germination rates of the transgenic lines and WT were counted after moving to normal conditions. OE-2 had the highest germination rate under cold treatment, OE-1 had the lowest germination rate among three transgenic lines, while the germination rates of all transgenic lines were significantly higher than WT (Figure 9I,J). After *Arabidopsis* reached the four-leaf stage, it was treated with cold for seven days and the root length was counted. As shown, the root lengths of the transgenic lines were significantly different from the WT, and the root number of three transgenic lines was also significantly different from the WT (Figure 9K–M). Moreover, the leaves of WT had turned to purple, while the transgenic lines remained green (Figure 9K). *Arabidopsis* was treated with cold for seven days when it approached the bolting stage. In the phenotype shown in Figure 9N, the WT rosette leaves virtually died, while the leaves of the transgenic lines still survived, especially in the OE-3 line (Figure 9N,O). The survival rate of transgenic lines was significantly higher than WT (Figure 9O), which was nearly three times higher than that of the WT. These results showed that transferring the *TaKNOX14-D* gene into *Arabidopsis* could improve the resistance to cold.

## 3. Discussion

The homeobox genes are essential and important in plant growth and development, and widely found in plants [33,34,35]. KNOX genes (KNOTTED1-like homeobox) belong to homeobox genes. Research has found that these genes affect the development of various organs by regulating the plant meristem [33,36,37]. However, there are few studies that have verified the KNOX genes’ functions in wheat [30,38,39]. Moreover, previous studies mainly focused on the effect of KNOX genes on plant development, and few research studies noticed the function of KNOX genes in response to abiotic stress [12,15,30,40]. In this study, the wheat KNOX genes were identified from the whole wheat genome, and the gene structures and gene expression patterns were analyzed. Meanwhile, the demonstration of the function of the KNOX genes in response to mechanism damage and cold stress would help to understand the KNOX genes’ functions in wheat.

This study identified 36 *TaKNOX* genes, which contained 34 *TaKNOX* genes identified by Han [30]. *TaKNOX* genes distributed in chromosomes unevenly. The numbers of genes on the three sub-genomes, A, B and D, were inconsistent, which might be the functional redundancy of *TaKNOX* genes in wheat evolution, resulting in some genes being lost in inheritance. The uneven distribution of *TaKNOX* genes on the chromosomes might be due to the specific retention and spread of *TaKNOX* genes during polyploidization. In addition, wheat also had more KNOX genes than the other monocot KNOX genes, that for the members of this gene family gradually increased with the evolution of multicellular diploid in plants [41,42]. Unicellular green algae and red algae contain only one KNOX gene, while in land plants, the KNOX gene family commonly consists of multiple members [24]. Between the 36 *TaKNOX* genes, there were 5 pairs of tandem repeats and 28 segmental duplications, suggesting that genome duplication events play an essential role in the amplification of the *TaKNOX* gene family in wheat. As a result, the *TaKNOX* gene family is larger than in *Arabidopsis* and rice. In addition, through analysis of the collinearity KNOX genes between rice, *Arabidopsis* and wheat, we found that more collinearity genes existed in wheat and rice, which might be because wheat and rice belong to monocot crops, while *Arabidopsis* belongs to dicot crops. This result was also verified by the phylogenetic tree of wheat and other plants. 

The miRNA is a regulator that plays an important role in the post-transcriptional regulation of the expression level of plant proteins [43]. Due to the number of wheat miRNA, which submitted to miRBase is few, there might be some *TaKNOX* genes not identified in the co-expression of miRNA. In this study, the miRNA of tae-miR408 was targeted by the most *TaKNOX* genes, because miRNA408 plays a role in plant photosynthesis and responded to much abiotic stress [44,45,46]. KNOX genes are important to plant leaf development and also respond to abiotic stress [2,47,48,49]. miRNA408 may regulate the chlorophyll content of plants by inhibiting the expression of the KNOX genes. 

Through analysis of the KNOX gene phylogenetic tree in various species including C3 and C4, results indicated that the KNOX genes were conserved during plant evolution. Moreover, *TaKNOX5-B*, *TaKNOX6-D1* and *TaKNOX8-A* were close to *STM,* (AT1G6230) and *KNAT1* (AT4G08150) in class I, which implied these three KNOX genes might have similar biological functions to *STM* and *KNAT1*, which affect the establishment of the SAM and the formation of embryonic callus in the embryo [10,21]. Thus, transferring these three KNOX genes into wheat might improve the efficiency of genetic modification. The genetic distance between the *TaKNOX14-D* gene and *KNAT3* and *KNAT7* genes was close, indicating that *TaKNOX14-D* functions might be similar to these two genes. Previous studies have proved that *KNAT3* and *KNAT7* are involved in the synthesis of monophenols in plants, and affected the deposition of secondary cell wall [27,50]. The wheat KNOX genes in the same group had similar gene structures, but the genes contained different numbers of introns. The number of introns is crucial for the evolution and production of plant gene families [51,52]. The promoter of the *TaKNOX* genes contained different *cis*-acting elements. Almost all genes had the *cis*-acting elements that respond to methyl jasmonate and abscisic acid, suggesting that the *TaKNOX* gene promoter was conserved and the *TaKNOX* gene might be involved in abiotic stress. Few KNOX also contained the *cis*-acting elements of GA response elements, suggesting that these KNOX might be involved in the elongation growth of plant stems.

The expression patterns of KNOX genes in wheat were divided into two groups, with seven *TaKNOX* genes located in class II, which were highly expressed during the wheat growth periods and highly expressed in all tissues. This group was also highly responsive to abiotic stress. The expression patterns of class I KNOX genes were concentrated, which were mainly expressed in the wheat SAM and stem, while not responsive to the wheat abiotic stress. These results were consistent with other studies [49,53,54]. Some KNOX genes in class I were highly expressed in the SAM and stem, especially in the early stem development, while the expression level was gradually decreased in later development. This result implied that these genes play an important role in the maintenance and regulation of the wheat SAM. The *TaKNOX* genes with up-regulated or down-regulated expression would help plant leaf development [54,55,56]. Our results of the *TaKNOX* expression pattern after mowing approved this view, and it also indicated that the KNOX genes might be involved in the compensatory growth of plants. Other studies have shown that the plant KNOX genes are important to plant de novo regeneration [53]. KNOX genes’ expression was regulated by hormones [7,57]. In this study, we found that the *TaKNOX* gene was up-regulated under ABA treatment, this result was similar to Han [30]. The expression of the *TaKNOX* gene was also regulated by CTK and IAA. Previous studies indicated that IAA suppressed the expression of KNOX gene, whereas CTK would promote KNOX gene expression [58]. However, in this study, the genes *TaKNOX2-A, 11-A* and *14-D* were up-regulated when treated with IAA, and CTK also promoted expression of these genes.

The KNOX gene has been proven to be involved in leaf development. In this study, the leaf area and chlorophyll contents of transgenic *Arabidopsis* lines were significantly larger than WT. GUS staining results showed the promoter of *TaKNOX* genes was expressed in leaves, which consists of the lager leaves of transgenic lines. *TaKNOX14-D* overexpression also resulted in the ratio of CTK and IAA being changed in leaves. When the ratio of CTK to IAA increased, it is beneficial to plant leaf development [59]. CTK could promote *KNOX1* expression during development in plants. Isopentenyl transfer (IPT) is a key gene in the CTK synthesis pathway [60]. Increasing IPT expression in tobacco results in the concentration of CTK increasing in plants and also increasing the expression level of *KNOX1* genes [61]. Previous studies also proved that increasing the expression levels of *KNOX1* in *Arabidopsis* significantly increased the concentration of CTK [58]. The root length of transgenic *Arabidopsis* was also longer than WT in this experiment. We speculated it was due to the seed sizes of transgenic lines being bigger than WT, which resulted in accumulation of more nutrition in seeds that was beneficial to root development. It might be the reason for the large leaf area of transgenic *Arabidopsis* as well. The seeds volume of the same crop is crucial for their germination rate and later growth [62,63]. The transgenic lines were higher than the WT line, presumably due to the higher chlorophyll content in the rosette leaf, which accumulated more photosynthetic products in the early stages. This result was consistent with the results of Yao [39]. *TaKNOX14-D* overexpression in *Arabidopsis* made the seed sizes bigger than WT, which was also consistent with the results of Yao et al. [39]. The regeneration speed of the transgenic lines after mechanical damage was faster than WT, which might be due to the initial accumulation of nutrients and the consequent acceleration of the plant regeneration process. There was no significant difference in seed size between the transgenic line and the WT after mechanical damage. However, the transgenic lines could produce more seeds, which also resulted in higher yields of the transgenic *Arabidopsis* than WT. The plant would redistribute nutrients and be preferentially supplied to regenerate injured tissue [64,65]. This may be responsible for the decrease of chlorophyll content in the rosette leaves after mechanical damage in *Arabidopsis*. The transgenic lines showed better resistance to cold stress, which was consistent with the gene expression pattern. Under cold stress, some wheat KNOX genes were up-regulated. The promoter region of the KNOX genes in wheat also had low-temperature responsive *cis*-acting elements, which also demonstrated that the KNOX genes might be involved in the cold stress response in plants. In addition, related studies have shown that the overexpression of KNOX genes in *Arabidopsis* could improve the drought resistance [30]. This study demonstrated that *TaKNOX14-D* in wheat could improve plant regeneration and improve cold tolerance in *Arabidopsis*, which also increased seed size in normal growing conditions. These results would help to understand the function of the KNOX genes in wheat.

## 4. Materials and Methods

### 4.1. Plant Materials and Treatment

Wheat line “XN 136” is an elite wheat variety, which was used in this study. The same size of wheat was selected and planted in a pot (20 cm bottom length * 25 cm height * 35 cm top length). Each pot contained 12.5 L of nutrient soil and eight wheat seeds. After the seeds were planted into the pot, thin soil was covered on the surface.

After the seeds germinated, five seedlings were kept in each pot. When the wheat seeding reached three leaves, the pot was moved into a pit in the field, and the mouth of the pot was kept at the same height as the ground. When the wheat grew 5–6 tillers, half of the wheats were selected for mowing, and stubble height for 2 cm was left. At 0 h, 0.5 h, 2 h, 8 h, 24 h, 72 h and 144 h after mowing, the root apical meristem and stem apical meristem of wheat were collected and mixed with liquid nitrogen. In this study, the materials were planted at the experimental station of Northwest A&F University, Yangling, China (34°20′ N, 108°24′ E).

For the hormone treatment experiment, the same size seeds were chosen and disinfected with 8% sodium hypochlorite for 24 h. After disinfecting, the wheat was washed with water 3–4 times. Wheat seeds were placed into the germination box evenly. One week after the seed germination, the wheat seedings were treated with different hormones for 1 h. After treatment, the whole wheat was collected and kept with liquid nitrogen. In this study, the hormones included abscisic acid (ABA), cytokinin (CTK), gibberellin (GA), auxin (IAA), jasmonic acid methyl ester (MeJA) and salicylic acid (SA), with concentrations according to Niu (2017) with some modification [66].

### 4.2. Identification and Bioinformatics Analyses of TaKNOX Genes

Wheat proteins were downloaded from the Ensembl Plant database (http://plants.ensembl.org/index.html (accessed on 22 September 2022)). Pfam v31.0 database (http://pfam.xfam.org/ (accessed on 24 September 2022)) was used to download KNOX1 and KNOX2 HMM profile (PF03790 and PF03791). An HMM profile was used to search against the wheat protein sequences by HMM search using a threshold E < 1 × 10^−5^ [67,68]. The adversarial results were submitted to the NCBI CDD to search the domain, removing the disqualified sequences (https://www.ncbi.nlm.nih.gov/cdd (accessed on 6 October 2022)) [69]. The KNOX protein sequences of rice, soybean and Arabidopsis were downloaded from the NCBI database (https://www.ncbi.nlm.nih.gov/ (accessed on 7 October 2022)). The protein sequences of families, maize, millet and tomato were also downloaded from Ensemble Plant and identified via HMM search [67]. The maximum-likelihood (ML) phylogenetic tree was constructed by MEGA X with 1000 bootstrap replicates, and the phylogenetic tree was optimized by EvolView (http:// www.evolgenius.info/evolview/ (accessed on 24 September 2022)) [70]. The subcellular localization of wheat proteins was predicted by Cell-PLoc 2.0 (http://www.csbio.sjtu.edu.cn/bioinf/Cell-PLoc-2/ (accessed on 26 September 2022)), and the physicochemical properties of proteins were predicted by Expasy (https://web.expasy.org/protparam/ (accessed on 10 October 2022)). 

### 4.3. TaKNOX Gene Duplication and Homology Analysis

Genes were mapped on the chromosomes by identifying their chromosomal positions provided in the wheat genome database. Gene duplication events of KNOX genes in wheat were investigated based on the following three criteria: (a) the alignment covered >80% of the longer gene; (b) the aligned region had an identity >80%; and (c) only one duplication event was counted for the tightly linked genes [71]. In order to visualize the duplicated regions in the wheat genome, lines were drawn between matching genes using the Circos-0.67 program (http://circos.ca/ (accessed on 18 October 2022)). Homology of the KNOX genes among rice, *Arabidopsis* and wheat were also analyzed and characterized by Tbtools [72].

### 4.4. Gene Structure, Conserved Motif and Cis-Acting Element Analyses

Gene sequences were obtained from the Ensemble Plant. The 2000 bp 5′ sequences upstream of KNOX genes were also downloaded from Ensemble Plant as the regulatory promoter regions. GSDS (http://gsds.cbi.pku.edu.cn/ (accessed on 12 October 2022)) was used to analyse the gene structure. The MEME suite web server (http://meme-suite.org/ (accessed on 14 October 2022)) was used to predicted protein conserved motifs. The *cis*-acting elements were analyzed using the Plant Care database (http://bioinformatics.psb.ugent.be/webtools/plantcare/html/ (accessed on 20 October 2022)) [73]. All above data were submitted to EvolView (http://www.evolgenius.info/evolview/ (accessed on 21 October 2022)) for prettified.

### 4.5. TaKNOX Genes and miRNA Co-Expression Networks Construction

*TaKNOX* genes cDNA sequences were submitted into the psRNATarget tool (https://www.zhaolab.org/psRNATarget/analysis?function=2 (accessed on 22 October 2022)) to search the target miRNA [74]. The regulatory network result of miRNA and TaKNOX genes was visualized through the cytoscape tool (http://www.cytoscape.org/ (accessed on 6 November 2022)).

### 4.6. Gene Expression Pattern of Tissues, Age Expression and under Different Abiotic Stresses

The expression patterns of KNOX genes in different tissues and at wheat growth periods were downloaded from Expvip (http://www.wheat-expression.com/ (accessed on 26 October 2022)). In this study, the tissue expression levels in roots, grain, leaf, stem, shoot apical meristem (SAM), root apical meristem (RAM), spike, awns, glumes, anther, stamen and pistil were analyzed. The wheat period at seeding stage, three leaf stage, fifth leaf stage and seven leaf stage, tillering stage, flag leaf stage, heading stage, anthesis, grain filling stage, milk grain stage and maturation stage were analyzed. All expression data were transferred with Log2. Tbtools was used to analyze the expression data [72]. 

### 4.7. RNA Isolated and qRT-PCR

In the field, when wheat reached the 5–6 tiller stage, the RAM and SAM were collected. The samples were pulverized with a mortar. Total RNA was extracted by using 1 mL RNAiso Plus* (TAKARA, Shiga-ke, Japan). The RNA was then spread in 1.0% agarose to detect the quality. The cDNA first strand synthesis capacity was 20 µL, which consisted of 2 µg RNA and 18 µL PrimeScript™ II 1st Strand cDNA Synthesis Kit (TAKARA, Shiga-ke, Japan). The real-time PCR was 10 µL, which contained 100 ng cDNA, 1 µL forward and reverse primer and TB Green^®^ Fast qPCR Mix (TAKARA, Dalian, China). qRT-PCR was repeated three times, performed in QuantStudio™ 5 Dx (Thermo Fisher Scientific, Waltham, MA, USA). The actin of the wheat genes (Gene ID: AB181991) was used as a control. The 2^−∆∆Ct^ method was used for fluorescence quantitative data analysis [75].

### 4.8. Target Gene and Promoter Cloned

Target gene sequence was downloaded from Ensemble Plant. Specific primers for gene and promoter cloning were designed via Oligo 7. In total, gene cloned mixture was 50 µL containing 2 µg cDNA and 2 µL forward and reverse primer and KOD mix (TOYOBO, Osaka, Japan), following the manufacture’s guidelines. The reaction solution was performed on a DNA amplification machine (Thermo Fisher Scientific, Waltham, MA, USA). PCR amplification procedures were as follows: initial denaturation at 98 °C for 30 s, followed by 35 cycles of denaturation at 98 °C for 10 s, annealing at 60 °C for 20 s, extension at 68 °C for 1 min, while the final extension was at 68 °C for 7 min. The PCR product was spread in 1.5% agarose and purification was done using the TIANgel Midi purification Kit (Tiangen, Beijing, China). The depurated product was connected to the pMD™18-T Vector (TAKARA, Shiga-ke, Japan). The following step was connecting the gene to the expression vector PCAMBIA1302 on *Nco I* site and connecting the promoter to PCAMBIA1391z on the BamH1 site by the ClonExpress MultiS One Step Cloning Kit (Vazyme, Nanjing, China).

### 4.9. Phonotype Analysis in Arabidopsis

After the target gene and promoter were cloned into pCAMBIA1302 and pCAMBIA1391z, the plasmid of these two expression vectors were extracted and transformed into agrobacterium GV3101. Then, the wild-type of *Arabidopsis thaliana* ecotype Columbia was infected with agrobacterium. *Arabidopsis* was cultured in a light temperature box and set for 16 h/light, 8 h/dark, with the temperature set to 24 °C. After *Arabidopsis* maturation, the transgenic *Arabidopsis* seeds were harvested as the T0 generation. The T0 seeds were then planted in 1/2 MS medium containing 50 µg/mL hygromycin and then transferred to soil. After Arabidopsis maturation, the seeds of the transgenic *Arabidopsis* were called the T1 generation. The same ways were used until the seeds of T3 were obtained. Genetic function validation was performed with the T3 generation transgenic *Arabidopsis thaliana*.

For gene function verification, T3 generation seeds and wild-type (WT) were cultured in MS medium. After *Arabidopsis* reached the four-leaf stage, the chlorophyll content of transgenic lines and WT was detected. The root lengths, leaf widths and lengths of four lines were measured by Image J (version: 1.8.0) and counted for 30 plants [76]. Likewise, the *Arabidopsis* were transplanted into vegetative soil for the next experiment. Comparison experiments between the transgenic lines and WT were then performed during the budding-flowering stage and the pod stage, respectively. The pod length and seed volume were counted after the transgenic lines and the WT pods ripened. Twenty pods close to the top of the stem were chosen, and twenty seeds were randomly chosen, and the seed width/length were counted for each line in this experiment. The GUS staining was performed using the X-gluc kit (Solarbio, BeiJing, China) when the *Arabidopsis* reached the pod period. Five tissues of *Arabidopsis* (root, stem, leaf, flower and pod) were used for the GUS straining experiment. The rosette leaves width and length of four lines were also measured. When four lines reached the bolting-flowering stage, the rosette leaves were collected. In this study, six *Arabidopsis* plants were randomly selected from each line, then the width and length of each plant’s rosette leaves was measured by Image J (version: 1.8.0). For error reduction, total leaf length and total leaf width of each *Arabidopsis* rosette leaf were counted in this study.

To verify the compensated growth capacity of the transgenic lines, the tissues above rosette leaves were cut with scissors at the bolting-flowering stage, and then the phenotypes of the transgenic lines and WT were investigated and counted after mechanical injury. The phenotypes were surveyed according to above.

For cold stress treatment, the transgenic lines and WT were planted in MS medium for one day, then a part of the MS medium was placed in a 4 °C refrigerator for one week. Seven days later, the *Arabidopsis* was put into a light temperature box for normal growth, the four lines germination rate was counted for six days and each line germination rate repeated three times. Furthermore, when *Arabidopsis* grew to the 4-leaf stage it was treated with low temperatures for a week again. The root lengths of the transgenic and wild-type lines were counted after one week. When the *Arabidopsis* was close to the pumping period under normal growth conditions, it was treated with low temperature for a week. After seven days, the *Arabidopsis* was moved to a light temperature box for normal growth, and the survival rate of four lines after cold treatment was counted.

### 4.10. Statistical Analysis

ImageJ software (version: 1.8.0, NIH, Stapleton, NY, USA) was used to measure the root length, seeds length/width and leaf width/length [76]. Excel 2019 (Microsoft Corporation, Lumia, WA, Redmond, USA) was used to analyze and construct the data. In all graphs, error bars indicating standard deviation and significant difference was indicated with “*” (*p* < 0.05) or “**” (*p* < 0.01).

## 5. Conclusions

In summary, the KNOX gene family was identified in this study, and the expression patterns under different stresses, wheat growth periods and wheat tissues were analyzed. *TaKNOX14-D* had high expression levels in wheat growth periods and in many wheat tissues, and this gene also responded to cold stress. Moreover, *TaKNOX14-D* was also highly expressed in the TN during the leaf regeneration stage. In addition, overexpression of *TaKNOX14-D* increased the leaf size, plant height and seed size in *Arabidopsis*, and also increased the regeneration speed after cutting. This gene also increased the *Arabidopsis* resistance to cold. These results might lead to a better understanding of the function of *KNOX* genes in plant growth and responses to abiotic stress

## Figures and Tables

**Figure 1 ijms-23-15918-f001:**
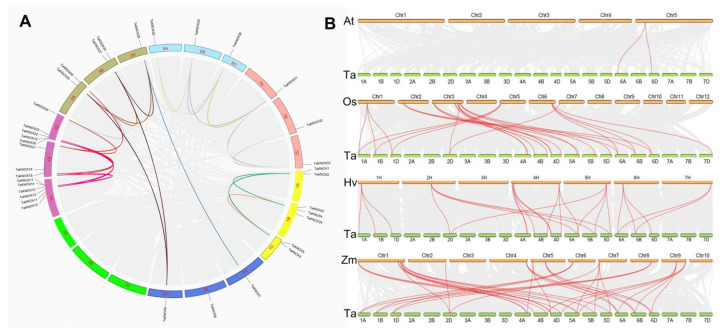
Distribution and duplication events of *TaKNOX* genes across the wheat genome. (**A**) *TaKNOX* genes are located at 18 chromosomes, and WGD/segmental and tandem duplications are mapped to their respective locations. The different colored curved frames represent the different chromosomes of wheat. Gray regions indicate all synteny blocks within the wheat genome, other color lines represent WGD/segmental and tandem duplications. The chromosome numbers are marked inside of the circle, each chromosome was represented by different colors. (**B**) Syntenic relationships of wheat KNOX genes among *Arabidopsis thaliana*, *Oryza sativa*. Genomic collinearity regions of wheat and other species are indicated by gray lines. The red lines indicate the collinear KNOX gene pairs. Ta: wheat, At: *Arabidopsis thaliana*, Os: rice, Hv: barley, Zm: maize.

**Figure 2 ijms-23-15918-f002:**
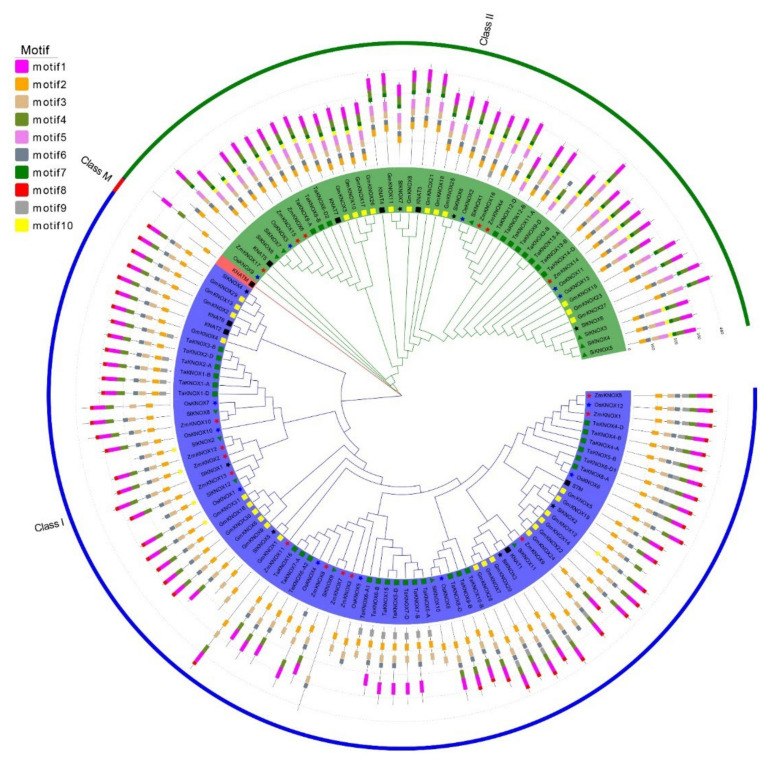
Phylogenetic tree of KNOX proteins constructed using the maximum-likelihood (ML) phylogenetic tree by MEGA X. In total, 126 KNOX genes from different species were used to construct the phylogenetic tree by MEGA X. The three different groups are indicated by different colors. Different species are indicated by different shapes and colors. 10 motifs are predicted by MEME and different colored boxes represent different motifs. The box lengths represent motif lengths.

**Figure 3 ijms-23-15918-f003:**
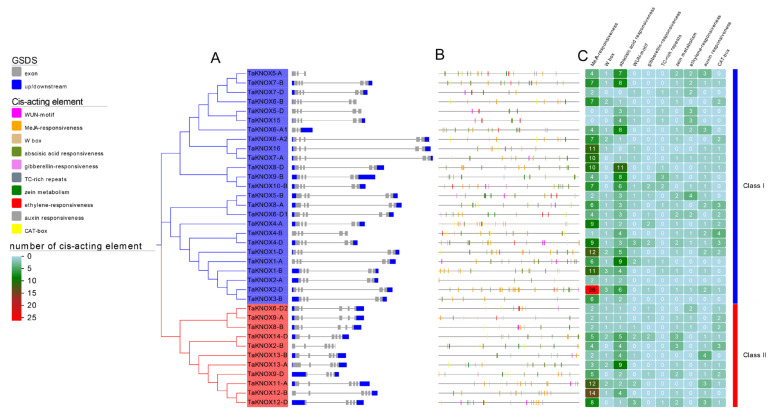
TaKNOX gene structures and *cis*-acting elements in TaKNOX promoters. (**A**) Phylogenetic tree of 36 *TaKNOX* genes and gene structures. Blue boxes represent UTRs, gray boxes represent exons and gray lines represent introns. (**B**) *cis*-acting element locations of TaKNOX gene promoters. The black line represents the promoter length, other different color lines located on the promoter represent the different *cis*-acting elements. (**C**) Number of *cis*-acting elements. Bule line and Red line represent the different KNOX groups.

**Figure 4 ijms-23-15918-f004:**
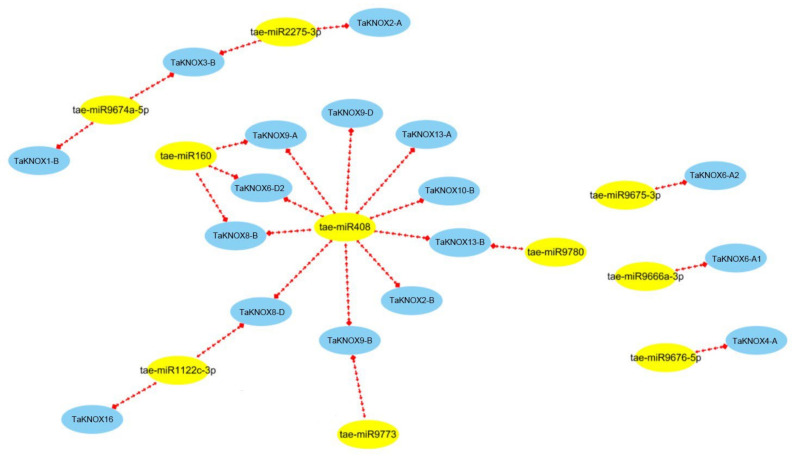
The co-expression regulatory network of TaKNOX cascade genes in wheat. Box color: blue, *TaKNOX* gene in wheat; yellow: miRNAs targeted by KNOX genes in wheat.

**Figure 5 ijms-23-15918-f005:**
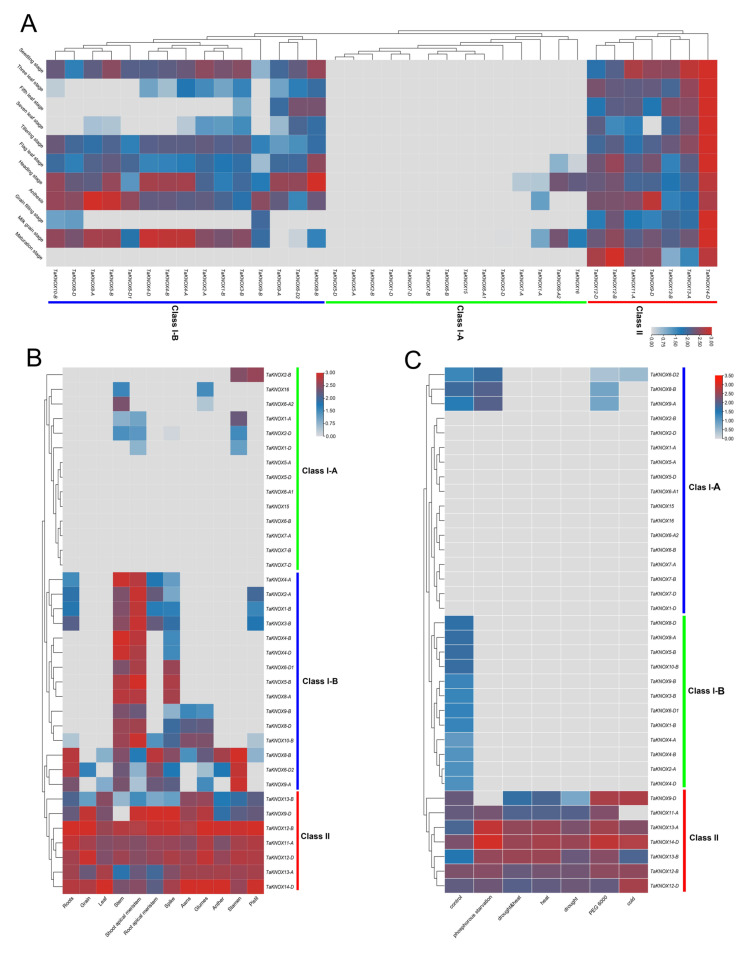
Heat map of the expression profiles of wheat TaKNOX genes at different wheat growth periods, different tissues and under different abiotic stresses. RNA-seq data of *TaKNOX* genes expression values of wheat growth period, tissues and under abiotic stresses were obtained from the wheat expression web. (**A**) The expression patterns of *TaKNOX* genes at different growth stages. (**B**) The expression pattern of *TaKNOX* genes in tissues. (**C**): The expression pattern of TaKNOX genes under different abiotic stresses. Different color lines represent different expression patterns.

**Figure 6 ijms-23-15918-f006:**
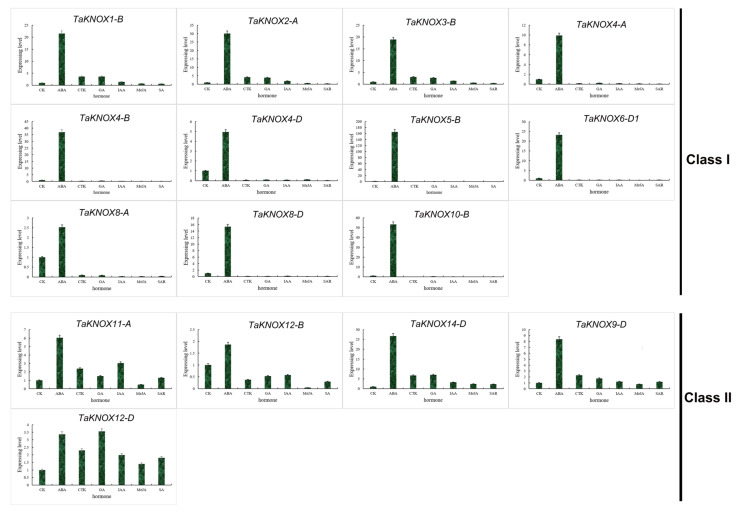
Expression patterns of 16 *TaKNOX* genes under hormone treatment obtained by qRT-PCR. The actin gene is used as an internal control. The data shown are means ± SD obtained from three biological replicates. ABA: abscisic acid, CTK: cytokinin, GA: gibberellin, IAA: auxin, MeJA: jasmonic acid methyl ester, SA: salicylic acid. Class I and class II represented different expression patterns.

**Figure 7 ijms-23-15918-f007:**
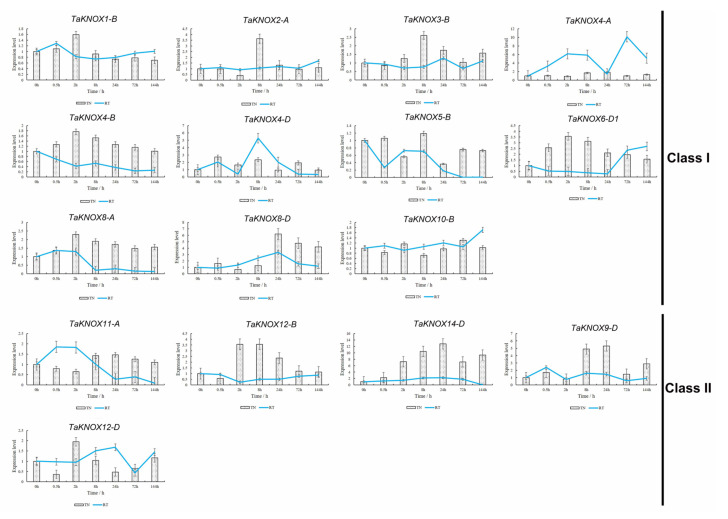
Expression patterns of 16 KNOX genes in response to mechanical treatment obtained by qRT-PCR. The actin gene is used as an internal control. The data shown are means ± SD obtained from three biological replicates. Histogram represents the *TaKNOX* genes expression pattern in wheat TN, line chart represents *TaKNOX* genes expression pattern in wheat RT. TN: tillering node; RT: root tip.

**Figure 8 ijms-23-15918-f008:**
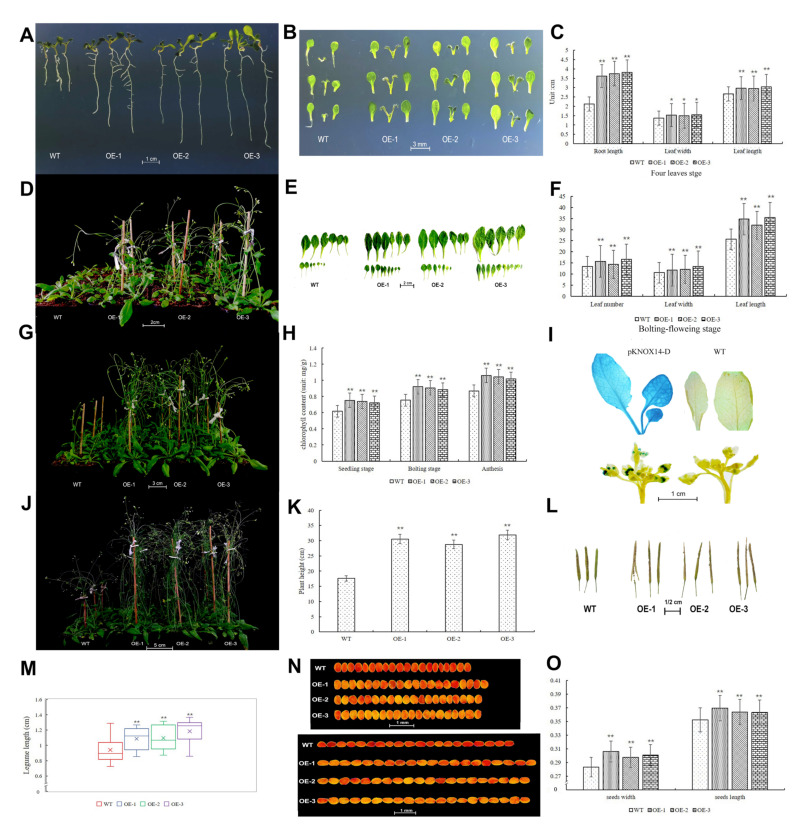
Phenotypes of WT and three transgenic *Arabidopsis* overexpressing *TaKNOX14-D*. (**A**) The root length of four *Arabidopsis* lines at the four-leaf stage, bar = 1 cm. (**B**) Leaf number and width/length at the four-leaf stage, bar = 3 mm. (**C**): Root length and leaf width and length of *Arabidopsis* at the four-leaf stage, 30 *Arabidopsis* plants were randomly selected from each line and the root length and leaf area were measured by Image J (version: 1.8.0). (**D**) Transgenic *Arabidopsis* at bolting stage and WT at seeding stage. (**E**,**F**) Leaf area and number of rosette leaves. Six *Arabidopsis* plants were randomly selected from each line and the leaf width and length were measured by Image J ((version: 1.8.0). (**G**) Transgenic lines at flowering stage and WT at bolting stage. (**H**) Chlorophyll content in rosette leaves, six plants of each line were randomly selected, the rosette leaves of six plants were detected with spectrophotometer and repeated three times. (**I**) GUS staining of a leaf and flower. (**J**) *Arabidopsis* at pod stage. (**K**) Plant height of four lines, six plants were randomly selected and plant height was measured with a ruler. (**L**,**M**) The pod length of four *Arabidopsis* lines, bar = 0.5 cm. Twenty pods located at the top of the Arabidopsis plant were collected and measured. (**N**,**O**) Seed length and width, bar = 1 mm. Twenty seeds were randomly chosen and the seed width/length were counted. *, *p* < 0.05; **, *p* < 0.01 (Student’s *t*-test).

**Figure 9 ijms-23-15918-f009:**
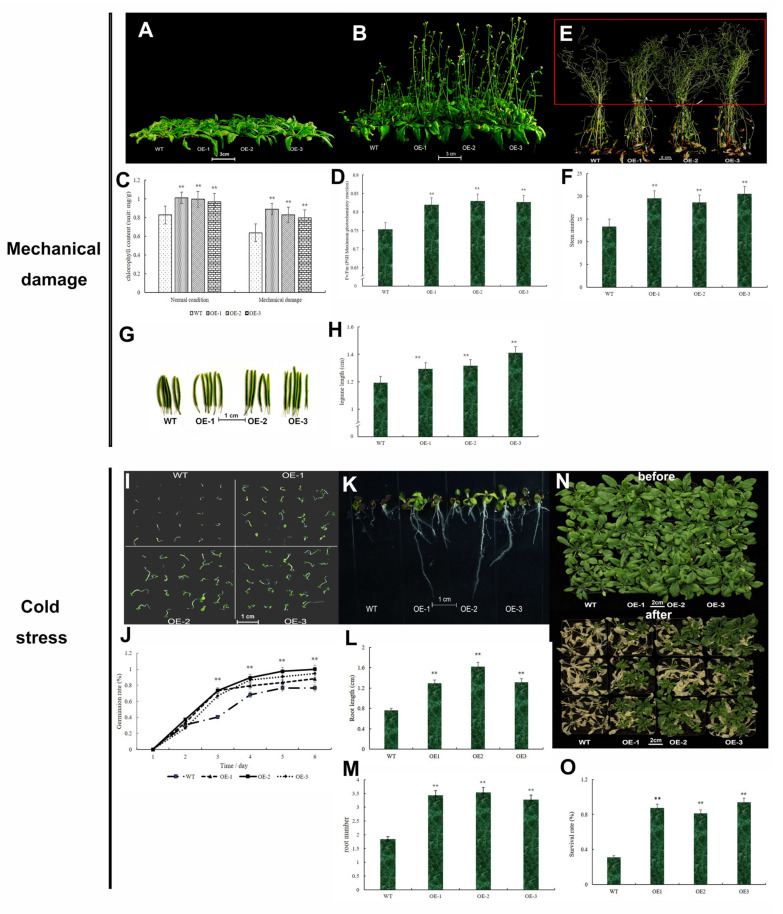
Phenotypes of WT and three transgenic lines after mechanical damage and under cold stresses. (**A**,**B**) Phenotypes of WT and three transgenic lines after mechanical damage. (**C**) Chlorophyll content in rosette leaves after mechanical damage. Six plants of each line were randomly selected, the rosette leaves of six plants were detected with spectrophotometer and repeated three times. (**D**) Fv/Fm after mechanical damage. Fv/Fm of twelve plants was detected in this study. (**E**) Four *Arabidopsis* at pod stage. (**F**) Stem number of four lines at pod period after mechanical damage. Six plant stems were counted. (**G**,**H**) Pod length, bar = 1 cm. Twenty pods located at the top of Arabidopsis plants were collected and measured. (**I**,**J**) The germination rate of four *Arabidopsis* lines after cold treatment. (**K**–**M**): Root length and number of four lines after cold treatment. The roots of twenty plants of each line were counted after cold treatment. (**N**,**O**) Phenotype of four lines after cold treatment and survival rate of four lines after cold treatment. Fifteen plants of each line were treated with cold for seven days, after treatment, the survival rate of each line was counted. **, *p* < 0.01 (Student’s *t*-test).

## Data Availability

Not applicable.

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
