# Peer review of "Genome-Wide Identification of Wheat KNOX Gene Family and Functional Characterization of TaKNOX14-D in Plants"

_ijms, 2022, doi:10.3390/ijms232415918_

Round 1
Reviewer 1 Report
This manuscript titled “Genome-wide identification of wheat KNOX gene family and functional characterization of the TaKNOX14-D in plants” presents essential new data about KNOX gene family. Materials and methods used in this study have been sufficiently described. It is worth emphasizing the authors' s analyses importance of both cis‑acting elements and miRNA target sites in KNOX genes expression.
Overall, this is an intriguing manuscript that presents the results of a well-planned and executed study. Before recommending this article for publication, there are several shortcomings for that should be resolve.
Some minor remarks:
Line 52: "mutant of as1and as2" should be "as1 and as2 mutants." and gene and mutant names should be italicized throughout the manuscript.
Line 56: “(Hay and Tsiantis, 2006; Shani et al., 2009)” Reference should be in same style like [11,31,39].
Line 68: "stm mutant" should be "STM mutant," and uppercase and lowercase should be consistent throughout the manuscript.
Line 70: “osh1 and osh15 double-deletion” should be “osh1/osh15 double-deletion”.
Line 119: It is more convenient to organize the chromosomes in order in TB-Tools, and you also performed this analysis in TB-Tools. Could you please arrange the chromosomes in Figure 1B in the correct order? Please also include the list of collinear KNOX gene pairs in the supplementary table.
Line 124: It would be preferable if you kept Figure 1A in the manuscript and added some more pairs of synteny in the supplementary figure, such as wheat-maize and wheat-barley.
Line 140: Could you please explain why class III is called class M? It is preferable to state the class name in a scientific manner (either letter or alphabet for all class).
Line 177-178: A recent study that looked at cis-elements with a 2000bp promoter could be cited. (Like: https://doi.org/10.3390/ijms23137187, https://doi.org/10.1007/s12374-020-09279-x)
Line 270 and 293: It is preferable to use two internal controls rather than one.
Line 90, 172, 176, 178, 479, 480, 482, 535: “cis-acting elements” should be “cis-acting elements”
Line 498: “de novo” should be written in italics “de novo”.
Line 568: “Hmm” should be capitalized “HMM”
Line 566, 574, 591, 592, 624: “ensemble plant” should be “Ensembl Plants”
Line 566 to 605: Webtool spontaneously modified their database. So, please add the access date of each webtool. Example: Ensembl Plants database (accessed on 6 June 2022).
Line 578: It's promising to see "Cell-PLoc 2.0" being used to predict localization. I believe that when compared to other webtools such as WoLF PSORT, PredSL, and so on, predicting subcellular localization will be more accurate. If possible, please include GFP localization to demonstrate that the prediction is valid.
Line 609: “mike grain stage” should be “milk grain stage”
Line 610: “Log2” should be “Log2”
Line 688: “Cod stress” should be “Cold stress”
Supplementary Table 1: It embodies the fundamental characteristics of this gene family. As a result, researchers can easily obtain their desired genes from this table. It should be included in the menuscript.
Supplementary Table 5: Cis-elements in the promoter region and gene exon location should be represented in different tables for reader convenience. In addition, the column title "Sequence" is not properly positioned.
Author Response
Line 52: "mutant of as1and as2" should be "as1 and as2 mutants." and gene and mutant names should be italicized throughout the manuscript.
Answer: Thank you for point out this mistake, we have corrected this error.
Line 56: “(Hay and Tsiantis, 2006; Shani et al., 2009)” Reference should be in same style like [11,31,39].
Answer: Thanks for point out this mistake, we have corrected this error.
Line 68: "stm mutant" should be "STM mutant," and uppercase and lowercase should be consistent throughout the manuscript.
Answer: Thanks for point out this mistake, we have checked and corrected this error.
Line 70: “osh1 and osh15 double-deletion” should be “osh1/osh15 double-deletion”.
Answer: Thanks for point out this mistake, we have corrected this error.
Line 119: It is more convenient to organize the chromosomes in order in TB-Tools, and you also performed this analysis in TB-Tools. Could you please arrange the chromosomes in Figure 1B in the correct order? Please also include the list of collinear KNOX gene pairs in the supplementary table.
Answer: We are sorry this mistake, we have corrected this. The list of collinear KNOX gene pairs was added in the supplementary table.
Line 124: It would be preferable if you kept Figure 1A in the manuscript and added some more pairs of synteny in the supplementary figure, such as wheat-maize and wheat-barley.
Answer: Thank your advice, we have added the wheat-maize and wheat-barley in the Fig.1, and we also rewrote the result.
Line 140: Could you please explain why class III is called class M? It is preferable to state the class name in a scientific manner (either letter or alphabet for all class).
Answer: Class M was only found in Arabidopsis, this group encodes a MEINOX domain but lacks the homeodomain, the researcher named the group according to the domain it encoded.
Line 177-178: A recent study that looked at cis-elements with a 2000bp promoter could be cited. (Like: https://doi.org/10.3390/ijms23137187, https://doi.org/10.1007/s12374-020-09279-x)
Answer: Thanks for point out this error, the reference of https://doi.org/10.3390/ijms23137187 was cited.
Line 270 and 293: It is preferable to use two internal controls rather than one.
Answer:
We are sorry for this confused you. Actually, there are two internal controls in this experiment. For a more intuitive understanding of the data, we normalized the expression data with TaActin gene, and set to a value of 1 of the expression level in TN and root under normal condition.
Line 90, 172, 176, 178, 479, 480, 482, 535: “cis-acting elements” should be “cis-acting elements”
Answer: Thanks for point out this mistake, we have checked and corrected this error.
Line 498: “de novo” should be written in italics “de novo”.
Answer: Thanks for point out this mistake, we have checked and corrected this error.
Line 568: “Hmm” should be capitalized “HMM”
Answer: Thanks for point out this mistake, we have checked and corrected this error.
Line 566, 574, 591, 592, 624: “ensemble plant” should be “Ensembl Plants”
Answer: Thanks for point out this mistake, we have checked and corrected this error.
Line 566 to 605: Webtool spontaneously modified their database. So, please add the access date of each webtool. Example: Ensembl Plants database (accessed on 6 June 2022).
Answer: Thanks for your advice, we have added the access date of each webtool in materials and methods section.
Line 578: It's promising to see "Cell-PLoc 2.0" being used to predict localization. I believe that when compared to other webtools such as WoLF PSORT, PredSL, and so on, predicting subcellular localization will be more accurate. If possible, please include GFP localization to demonstrate that the prediction is valid.
Answer: Thank you for your advice. Your suggestion is helpful to further modify and improve this manuscript. But this experiment can’t finish in a short time, the GFP localization will be reflected in follow-up study.
Line 609: “mike grain stage” should be “milk grain stage”
Answer: Thanks for point out this mistake, we have checked and corrected this error.
Line 610: “Log2” should be “Log2”
Answer: Thanks for point out this mistake, we have checked and corrected this error.
Line 688: “Cod stress” should be “Cold stress”
Answer: Thanks for point out this mistake, we have checked and corrected this error.
Supplementary Table 1: It embodies the fundamental characteristics of this gene family. As a result, researchers can easily obtain their desired genes from this table. It should be included in the manuscript.
Answer: Thanks for point out this mistake, we have checked and corrected this error.
Supplementary Table 5: Cis-elements in the promoter region and gene exon location should be represented in different tables for reader convenience. In addition, the column title "Sequence" is not properly positioned.
Answer: Thanks for point out this mistake, we have checked and corrected this error.
Reviewer 2 Report
The current study is interesting and merit for consideration with IJMS-MDPI. I have few minor point for consideration of authors.
Modify the following sentence, "TaKNOX14-D This study provides better under- 25 standing of the TaKNOX genes".
Check the gene/s name, they must be italic.
Better to mention the scientific names of crop species e.g., Wheat, Tomato etc., rather their generic names.
Figure 6 & 7 resolution is too low, improve the resolution.
Avoid the discussion in result section.
Remove the outdated citations.
Author Response
Modify the following sentence, "TaKNOX14-D This study provides better under- 25 standing of the TaKNOX genes".
Answer: Thanks for point out this mistake, we have checked and corrected this error.
Check the gene/s name, they must be italic.
Answer: Thanks for point out this mistake, we have checked and corrected this error.
Figure 6 & 7 resolution is too low, improve the resolution.
Answer: Thanks for your advice, we have improved resolution of figure 6 & 7.
Avoid the discussion in result section.
Answer: Thanks for your advice, we have checked result section and removed the discussion.
Remove the outdated citations.
Answer: Thanks for point out this mistake, we have removed the outdated citations and citated new reference.